# Milk Processing Affects Structure, Bioavailability and Immunogenicity of β-lactoglobulin

**DOI:** 10.3390/foods9070874

**Published:** 2020-07-03

**Authors:** Kerensa Broersen

**Affiliations:** Department of Applied Stem Cell Technologies, TechMed Centre, University of Twente, Postbus 217, 7500 AE Enschede, The Netherlands; k.broersen@utwente.nl

**Keywords:** bovine milk, pasteurization, β-lactoglobulin, digestion, aggregation, lactosylation, gastro-intestinal immune system

## Abstract

Bovine milk is subjected to various processing steps to warrant constant quality and consumer safety. One of these steps is pasteurization, which involves the exposure of liquid milk to a high temperature for a limited amount of time. While such heating effectively ameliorates consumer safety concerns mediated by pathogenic bacteria, these conditions also have an impact on one of the main nutritional whey constituents of milk, the protein β-lactoglobulin. As a function of heating, β-lactoglobulin was shown to become increasingly prone to denaturation, aggregation, and lactose conjugation. This review discusses the implications of such heat-induced modifications on digestion and adsorption in the gastro-intestinal tract, and the responses these conformations elicit from the gastro-intestinal immune system.

## 1. Introduction

Milk of bovine origin is widely consumed at a global level. While bovine milk is a protein-rich beverage containing 30–36 g of total protein per liter, the precise composition of milk varies with animal breed, stage of lactation, age, and diet (reviewed in [1]). The most prominent proteins present in milk are the caseins consisting of α-s1, α-s2, and β and κ-caseins, and the whey proteins α-lactalbumin and β-lactoglobulin. While the caseins are dispersed in the form of calcium-containing micelles, whey proteins are solubilized in the whey fraction. Apart from a rich source of proteins, bovine milk also contains the sugar lactose, an O-β-D-galactopyranosyl-(1→4)-D-glucopyranose, and lipids, primarily in the form of triacylglycerols. Prior to consumption, raw milk directly derived from the animal is subjected to various processing steps that may affect its molecular constituents. This review describes the available literature investigating the effect of processing of bovine milk on the structure and bioavailability of the main whey protein β-lactoglobulin.

## 2. β-Lactoglobulin Secretion, Structure, and Genetic Variants

### 2.1. β-Lactoglobulin is a Member of the Lipocalin Family

β-Lactoglobulin is the predominant whey protein in bovine milk. After β-lactoglobulin was recognized to share structural similarities [2,3] and homology [4,5] with plasma retinol binding protein, both proteins were assigned to the extracellular lipocalin family [5,6], together with mouse major urinary protein [7], insecticyanin [8], and α_2u_-globulin [9]. The lipocalin family, in turn, is part of the structural superfamily of calycins that, apart from lipocalins, also includes fatty acid-binding proteins, bacterial metalloprotease inhibitors, avidins, and triabin, which is a serine protease inhibitor. Although the biological function of β-lactoglobulin remains a debated topic, lipocalins share structural motifs giving rise to the ability of β-lactoglobulin to bind to small hydrophobic ligands such as fatty acids, cholesterol, vitamin D_2_, or carotenoids within its hydrophobic cavity [10,11] in addition to a possible weak binding site elsewhere on the molecule [12] that is genetic variant-dependent and becomes exposed upon heat-induced denaturation [13]. Transport of small hydrophobic ligands has therefore been suggested as a potential biological role of β-lactoglobulin. β-Lactoglobulin naturally occurs as a non-covalently bound dimer and this dimerization was found to play a critical role determining the affinity of β-lactoglobulin to ligands [14]. Recently, the natural affinity of β-lactoglobulin for fatty acids and other hydrophobic ligands initiated its exploitation to facilitate its potential as a drug carrier [15,16,17]. Other suggested biological roles of β-lactoglobulin include neonatal passive immunity transfer or a role in metabolism of phosphate in the mammary gland as a result of its observed interaction with p-nitrophenyl phosphate [18].

### 2.2. β-Lactoglobulin is Post-Translationally Modified and Secreted

β-Lactoglobulin is produced in the epithelial cells of the mammary gland. Pre-β-lactoglobulin is synthesized on membrane-bound polysomes [19] and contains a highly conserved signal peptide [20,21], destining this protein for its secretion. Upon trafficking from the ER to the Golgi for continued processing, pre-β-lactoglobulin comes across the sugar lactose which is generated in the trans-Golgi complex. Water buffalo β-lactoglobulin has also been found to be subject to N-acylation or N-lauroylation [22] although these modifications have not been reported for bovine β-lactoglobulin. Milk is secreted from the bovine mammary gland upon fusion of lactose and milk protein-containing secretory vesicles with the apical plasma membrane [23]. Secretion of β-lactoglobulin from the mammary gland is regulated by the peptide hormone prolactin, of which the production is triggered in response to neuroendocrine reflex, and supported by other hormones such as insulin, cortisol, thyroid hormone, and oxytocin which acts on contractile cells surrounding the ducts and alveoli [24]. Prolactin signaling involves activation of a Janus kinase (JAK)-signal transducer and activator of transcription proteins (STAT) pathway inducing transcription of genes involved in lactogenesis (reviewed [25]).

### 2.3. β-Lactoglobulin Fold is a β-Sheet Rich, Non-Covalently Coupled Homodimer

Upon the publication of an isolation procedure of β-lactoglobulin from bovine milk [26], structural and physico-chemical studies ensued to identify the structural characteristics of this protein. These studies showed that β-lactoglobulin is an 18.3 kDa, 162 amino acid polypeptide (Figure 1) which folds into an eight-stranded antiparallel β-barrel, arranged into two groups of four strands, lined with hydrophobic residues to create a deep hydrophobic cavity with ligand binding affinity [10,11]. A three-turn α-helix is attached to the outside of the barrel. A ninth β-strand lies at the exterior of the β-barrel and is involved in dimer formation [27]. The native molecule contains two intramolecular disulfide bonds (Cys106-Cys119 and Cys66-Cys160) and one free thiol group (Cys121). The free thiol group, Cys121, and the disulfide bond Cys106-Cys 119, are buried, whereas the other disulfide bond, Cys66-Cys160, is positioned on the outer surface in a mobile region of the molecule [3,28]. Between pH 5.5 and 7.5, at room temperature, β-lactoglobulin mainly exists as a non-covalently linked dimer. By increasing the temperature, addition of SDS, or moving the pH further away from the isoelectric point of the protein, the dimer–monomer equilibrium is shifted towards the monomeric form [29].

### 2.4. β-Lactoglobulin Knows as Many as Ten Genetic Variants

Some ten variants of bovine β-lactoglobulin have been identified, dependent on the animal breed, but the two major genetic variants are designated as A and B. The primary structures of these variants differ at residues 64, at which an aspartic acid in variant A is a glycine in variant B (A/B: D64G), and 118, at which the valine in variant A is an alanine in variant B (A/B: V118A) [32] (Figure 1). These two amino acid variations give rise to small differences in molecular weight and isoelectric point [33,34]. Even though this difference is minor, it has been shown to account for differences in stability upon exposure to heat [13,35,36,37], ligand binding [13], and susceptibility to proteolysis [38]. For example, bovine β-lactoglobulin variants B and C, but not variant A, were observed to form high molecular weight aggregates at pH 6.7 [37], and variant B was found to be more thermostable than variant A [13].

## 3. Processing of Milk on β-Lactoglobulin Folding and Structure

Upon collection of milk from a dairy farm the raw milk is subjected to a standardization step, by means of creaming, to generate a product with a defined level of fat. Creaming is generally performed at a temperature of 40 °C using centrifugal force to separate the skim milk from the cream. Subsequently, the milk is standardized to a predefined fat level by recombining the skim milk and cream. The standardized milk is then treated to extend the shelf life by heat-mediated removal of pathogenic microorganisms. While specific heating conditions may vary between dairy suppliers and countries, minimal heating conditions have been defined by the heat resistance of the pathogen *Coxiella burnetii*, which was identified in 1950 in an epidemiological study relating raw milk consumption to the occurrence of Q fever [39]. For example, the process of pasteurization may involve heating at 85 °C for 2 to 3 s or maintaining the milk at a temperature of 63–66 °C for 30 min. Sterilization is performed at higher temperatures of 107–130 °C; the milk is maintained at this temperature for 8 min up to 40 min. Other types of heating raw milk may be subjected to ultra high temperature treatment, which involves either steam injection or indirect heating at temperatures up to 145 °C for approximately 4 s [40]. Naturally, these processing conditions will impact the heat-sensitive biomacromolecules that milk is made up of. The stages of liquid milk processing and a summary of the experimentally-validated modifications in β-lactoglobulin as a consequence of processing are summarized in Figure 2.

### 3.1. Impact of Heat-Processing on β-Lactoglobulin Structure

#### 3.1.1. Denaturation and Molten Globule State

Bovine β-lactoglobulin is a globular protein that undergoes conformational changes as a function of pH or heat. While the native non-covalently bound dimeric structure of β-lactoglobulin is thermodynamically the most stable state in unprocessed milk with a small fraction occupying metastable partially unfolded monomeric states, heating and other types of processing and conditions can reportedly shift this relationship [41,42,43,44] as a result of dissociation of the hydrogen bond network [45]. For example, using sedimentation velocity analysis it was reported that β-lactoglobulin acquires monomer–dimer equilibrium at 100 mM NaCl and pH 2.5 [41]. Initially β-lactoglobulin conformational shifts upon heating involve destabilization and partial unfolding of its globular structure to expose histidine, tyrosine, and tryptophan residues to the solvent [46] and increasing reactivity of the buried thiol group [47]. These limited and reversible conformational changes are followed by dissociation of intramolecular interactions and occupation of a partially unfolded, so-called molten globule state [48], reviewed in [49], which is also populated upon refolding of β-lactoglobulin [50], yielding exposure of the thiol group and the buried hydrophobic core of the protein [51]. While initial conformational transitions occur at temperatures as low as 40 °C [52], fully denatured β-lactoglobulin is only observed at temperatures exceeding 130 °C [53,54,55], indicating that the denaturation process of β-lactoglobulin can be described as a multistep mechanism. Generally, the midpoint of transition, at which abrupt large-scale loss of α-helix structural elements is observed, is approximately 65 °C [55], depending on environmental conditions.

#### 3.1.2. Partially Denatured β-Lactoglobulin is Prone to Aggregation

The partially denatured β-lactoglobulin protein molecule has been widely reported to be prone to thiol/disulfide exchange reactions, leading to its covalent aggregation, with non-covalent interactions playing a minor role [56,57,58,59,60,61,62,63,64]. A quantitative kinetic model that describes the irreversible aggregation of β-lactoglobulin as a function of thiol/disulfide exchange reactions was reported under low salt, neutral pH conditions upon heating at a temperature of 60–75 °C [64]. Based on radical polymerization reactions, initiation, propagation, and termination steps were defined, translating into exposure of a free sulfhydryl group, a thiol/disulfide exchange reaction, and the reaction of two reactive intermediates, respectively. Studies like these have been highly instrumental in driving forward our understanding of the denaturation and aggregation of β-lactoglobulin in test tube conditions, which, however, in many cases do not resemble the complex environment in the presence of lipids, lactose, and other proteins and conditions in milk. For example, it has been shown that β-lactoglobulin can interact with other milk constituents, such as α-lactalbumin, caseins, and bovine serum albumin, lactose, and lipids, which remarkably changes β-lactoglobulin behavior. The impacts of such interactions on β-lactoglobulin behavior are discussed in Section 5.

### 3.2. Homogenization Induces Disruption of the Milk Fat Globule Membrane

Upon secretion of cytoplasmic milk fat globules, the vesicle is surrounded by an additional layer of plasma membrane, resulting in a phospholipid trilayer of which the outer layer is associated with plasma membrane-derived proteins and other biomolecular constituents termed the milk fat globule membrane (MFGM). Milk lipid globules vary in size from 0.1 to 20 µm and are secreted by lactocytes, specialized epithelial cells residing in the mammary alveolus. Creaming is the process of gravity separation of these milk fat globules, or cream, from the skim milk observed in unhomogenized milk. To prevent creaming, homogenization is used; it acts by disrupting the MFGM and reducing the size of these lipid globules to an average of 1 µm, thereby reducing their tendency to cause creaming [65,66]. The process of homogenization generally involves heating of the raw milk to a temperature of 40 °C followed by passing of the milk through a small-pore sized opening. While some studies show negligible effects of homogenization on structure and digestibility of β-lactoglobulin [67], one study showed that homogenization affects intestinal pancreatin-mediated digestion of raw and pasteurized whole milk. It was suggested that the MFGM-disrupting action of homogenization allows interaction with milk constituent proteins, resulting in their partial resistance against the enzymes trypsin and chymotrypsin [68]. In line with this suggestion, β-lactoglobulin was shown to interact with MFGM proteins but both homogenization and heat processing are prerequisites for this binding to occur [66]. The identified disulfide bond-mediated interaction between MFGM proteins and β-lactoglobulin supports the idea that cysteine residues that are largely buried within the hydrophobic core of the native protein require exposure by means of heat-induced unfolding [65]. Upon intestinal digestion, the bile salts are considered to replace proteins and peptides from the surface of milk lipid globules [69]. The interaction of β-lactoglobulin with MFGM proteins will be further discussed in Section 5.

### 3.3. Amino Acid Modifications

#### 3.3.1. The Maillard Reaction

Heat exposure of proteins in the presence of carbonyl compounds, such as reducing sugars, induces a complex non-enzymatic browning reaction termed the Maillard reaction [70]. In this reaction a reducing sugar targets the ε-amino group of lysine or the N-terminal group of a protein to form Amadori or Heyn’s rearrangement products. As originally described by Hodge in 1953 [71], the advanced stages of the Maillard reaction involve degradation of the Amadori (1-amino-1-deoxy-2-ketose) and Heyn’s products via a range of pathways, depending on the conditions under which the reaction takes place, involving Schiff bases, Strecker degradation, or fission products, ultimately giving rise to melanoidins, brown nitrogenous polymers, and copolymers thereof [72,73], thereby affecting sensory properties of the processed milk.

#### 3.3.2. Heating Affects the Degree of Lactose Conjugation to β-Lactoglobulin

Lactose is a reducing sugar present in milk that was shown, using mass spectrometry analysis, to conjugate to β-lactoglobulin upon heating by means of the Maillard reaction [74,75] reducing the number of available free lysine residues. While β-lactoglobulin in raw milk was found to also be lactosylated and even dilactosylated to some extent [76], processing of milk by means of heat treatment was shown to increase lactose conjugation. The degree of lysine modification depends on the intensity of the heat treatment [76]: longer incubation times [77] at higher temperatures result in more extensive conjugation of β-lactoglobulin, and vice versa. Using reverse phase liquid chromatography–mass spectrometry, or LC–MS, it was found that the pattern of lactosylation of β-lactoglobulin is indicative of the extent of the heat treatment milk has been subjected to [78]. Many of the lactosylation sites of β-lactoglobulin in commercially available pasteurized milk have now been identified and include K8, K14, K47 [74,79,80], K60, K69, K75, K77, K83, K91 [79,80,81], K100 [74], K101, and K135 [75,76]. At the same time, the complexity of the reaction dictates that each individual reaction is regulated differentially at different temperatures and alternative degradation pathways of the formed intermediates prevail in a temperature-dependent manner [82], resulting in a heterogeneous reaction mixture of β-lactoglobulin conjugated forms.

#### 3.3.3. Effects of β-Lactoglobulin Glycation on Structure and Aggregation

Glycosylation affects protein thermal stability [83,84,85,86]. Upon low degrees of conjugation of glucose or fructose to β-lactoglobulin, we and others observed that, while β-lactoglobulin became more resistant to denaturation upon heating [87,88,89,90], glucose conjugation resulted in loss of stability under conditions of chemical-induced unfolding using urea [91]. These two seemingly contradicting results were later reconciled in a publication which showed that glucose conjugation to β-lactoglobulin lowered the heat capacity change of unfolding [92]. In extension to this observation, conjugation of β-lactoglobulin with α-dicarbonyl compounds, a Maillard reaction intermediate [93], or glucose [94], was shown to inhibit fibril growth. At the same time, published observations showed that denaturation and initial stages of aggregation of β-lactoglobulin may actually be accelerated upon conjugation with reducing sugars, while the elongation of amyloid-like fibrils was reduced by inhibition of disulfide bond formation [95], increased steric hindrance [96,97], or disruption of hydrophobic interactions favoring aggregation [97]. Differences in degrees and site of reducing sugar conjugation to β-lactoglobulin and the Maillard reaction conditions may well explain these seemingly contradictory results by affecting native β-lactoglobulin conformation to a different extent. In a study investigating the aggregation—at an elevated temperature and pH 2—of β-lactoglobulin as a function of glucose or lactose conjugation, different lysine residues were shown to vary in their susceptibility to glycation to either reducing sugar which, in turn, differentially affected the different stages of the aggregation process [97].

## 4. A Special Type of Fold: β-Sheet Motif Spherical Particles and Amyloids

Self-assembly states or aggregates of β-lactoglobulin can be formed upon heat-denaturation and refolding concurrent with the formation of non-native intermolecular interactions. The ability to form such aggregates is both genetically variant and pH-dependent [37]. Aggregating proteins were first recognized in the light of a pathogenic species involved in Alzheimer’s disease, transmissible spongiform encephalopathies, Creutzfeldt-Jakob disease, type II diabetes, Parkinson’s disease, and Huntington’s disease [98,99,100,101,102]. Later studies showed that aggregate formation is not limited to the proteins related to these pathologies but that the ability of proteins to adopt an aggregated state is a shared feature of proteins independent of their amino acid sequences [103], reviewed in [104]. Additionally, while proteins are known to resume their fully folded state upon removal of heat, partial unfolding of β-lactoglobulin upon heating may expose reactive groups kinetically driving the formation of intermolecular interactions. The fate of a protein’s folding upon removal of heat then depends on the relationship between the rates of aggregation and refolding. Refolding of partially unfolded protein molecules into intermolecular cross-β-sheet structures has been central to one of the hypotheses by which β-lactoglobulin is thought to self-assemble into aggregates [105]. The concentration dependence of the aggregation rate than dictates the outcome of heating followed by cooling [106]. Additionally, while β-lactoglobulin concentrations may not be very high in fluid milk, it is likely that β-lactoglobulin may encounter other proteins that induce its co-aggregation (see also Section 5). Other than protein concentration, the extent of aggregation and the aggregate morphology are dependent on hydrophobicity, duration of heat treatment, and electrostatics [107]. While β-lactoglobulin was shown to populate two distinct aggregate morphologies as a function of pH, the fibrils formed are heterogeneous [108]. Near the iso-electric point of 5.2 and a transition temperature of 70–90 °C, the aggregate morphology of β-lactoglobulin is dominated by 100–1500 nm radii spherical aggregates with low β-sheet content [109,110,111]. On the other hand, fibrillar cross β-sheet-rich motif morphologies are found for β-lactoglobulin that has been incubated at a pH of 2, far removed from its iso-electric point [109,110,111,112]. These observations indicate that, even though the driving force for aggregation is ultimately overcome, electrostatics and charge density determine the morphological outcome of the reaction. Furthermore, at pH 2 it was shown that β-lactoglobulin amyloid fibrils are not composed of full-length monomers of β-lactoglobulin, but instead that hydrolysis-mediated peptide fragments self-assemble into oligomers that further elongate into fibrils [113]. A review summarizing the different morphological appearances of β-lactoglobulin aggregates and their molecular make-up has been published before [114]. Many insights on β-lactoglobulin aggregation have been rendered using in vitro studies making use of purified forms of β-lactoglobulin. In milk upon pasteurization also, aggregation of β-lactoglobulin [115,116] into large > 670 kDa assemblies [116] has been reported. For example, while aggregated β-lactoglobulin in raw milk was not detected, processed milk led to loss of native fold β-lactoglobulin concurrent with conversion into aggregates, as detected by native and SDS-PAGE depending on the processing conditions [115].

## 5. Interaction of β-Lactoglobulin with other Milk Proteins

Mixtures of proteins isolated from milk have been shown to interact in test tube conditions upon heating. For example, the presence of β-lactoglobulin induced the aggregation of α-lactalbumin, presumably by interaction at the unfolded state, to form mixed protein gels [117]. In skim milk, β-lactoglobulin was found to interact with casein micelles [118], κ-casein [119,120], α-lactalbumin [121], or MFGM proteins [65,66].

### 5.1. Interaction of β-Lactoglobulin with α-Lactalbumin

Besides β-lactoglobulin, α-lactalbumin is another major whey protein component in bovine milk containing four disulfide bonds but no free thiol groups [122]. As a regulatory subunit of the enzyme lactose synthase, α-lactalbumin acts by regulating the conjugation of glucose to the enzyme β1,4-galactosyltransferase mediating the formation of lactose [123]. Depending on the presence of calcium, α-lactalbumin assumes a moderately stable holo or native calcium-bound or unstable apo calcium-free conformation [124,125]. Compared to β-lactoglobulin, α-lactalbumin has demonstrated some resistance to the formation of aggregates. The lack of free thiol in α-lactalbumin to form intermolecular disulfide bonded aggregates has been proposed to contribute to this behavior and this can be circumvented by the exposure of β-lactoglobulin and α-lactalbumin mixtures to high pressure [126] or oil/water interface [127,128] conditions in which the free thiol group of β-lactoglobulin acts as a triggering agent forming mixed disulfide bonded β-lactoglobulin/α-lactalbumin oligomers [126]. A certain degree of temperature dependency of α-lactalbumin/β-lactoglobulin co-aggregation has been reported. For example, while temperature treatment of 72 °C for 15 s did not show formation of SDS-stable interactions between α-lactalbumin and β-lactoglobulin [129], exposure to high temperature led to detection of α-lactalbumin and β-lactoglobulin mixed aggregates stabilized by means of disulfide bond interchange [129,130,131,132] and hydrophobic interactions [132]. Extensive aggregation can develop into a gel state, which is a three-dimensional arrangement of molecules with water holding capacity. β-Lactoglobulin is capable of forming gels and the ability to do so at pH 2 was found to be induced by the presence of α-lactalbumin [133].

### 5.2. Interaction of β-Lactoglobulin with Caseins

Caseins in milk are generally found assembled into large structures termed casein micelles. One of the caseins in bovine milk is κ-casein, which is an amphiphilic protein associated with the surfaces of casein micelles by means of hydrophobic interactions in bovine milk masking the cysteine residues, while κ-casein hydrophilic regions extend away from the micelle and play a micelle stabilizing role [134,135]. After the original discovery that κ-casein and β-lactoglobulin form complexes upon heating of milk [136], in some cases also found to involve α-lactalbumin [137,138,139,140], further studies showed that this interaction is mainly disulfide bond driven [141]. Investigations on the specific residues involved in the reaction identified that cysteine 121 β-lactoglobulin interacts with cysteine 88 of κ-casein upon heat treatment, although other cysteine residues were observed to be involved as well, depending on the incubation conditions [142,143,144,145]. Casein micelles were found to either dissociate upon interaction with β-lactoglobulin [135] or to precede interaction with denatured β-lactoglobulin [146]. In turn, an increased rate of native protein conversion into small aggregates with inhibited progression into larger aggregates was reported for κ-casein-associated β-lactoglobulin [145] with a heterogeneous range of aggregated complexes being formed [120,141,142,143]. There has been some reported inconsistency in the aggregate size of β-lactoglobulin to be involved in the complexation with κ-casein, with some reporting that aggregation of β-lactoglobulin is a prerequisite for interaction with κ-casein [141,147], while others suggest that denatured [148] or small aggregated [149] forms of β-lactoglobulin are required for this interaction. It would make sense intuitively to suggest that the β-lactoglobulin fold at least requires some degree of unfolding to expose the thiol group that was previously identified to play a role in the disulfide bond coupling of β-lactoglobulin to κ-casein. Although many of the studies investigating the interaction between κ-casein and β-lactoglobulin were based on purified or partially purified proteins, it was shown that in heated bovine milk also, similar patterns for complexation were identified [150]. A review further detailing the heat-induced interaction between whey protein fragments and κ-casein has been published before [151].

### 5.3. Interaction of β-Lactoglobulin with MFGM Proteins

MFGM proteins consist of various proteins derived from the plasma membrane of the secretory epithelial cells of the mammary gland. This origin results in a complex mixture of major proteins entailing the glycoprotein lactadherin (periodic acid Schiff (PAS) 6/7), xanthine dehydrogenase/oxidase, periodic acid Schiff III, acidophilin, mucin MUC1, butyrophilin, fatty-acid binding protein (FABP), and cluster of differentiation (CD) 36, in addition to minor fractions of enzymes, cytoplasmic proteins, immunoglobulins, major histocompatibility complex (MHC) molecules (reviewed in [152]), and apolipoproteins [153]. β-Lactoglobulin was reported to be associated with MFGMs by covalent disulfide bond interaction in a few occasions [153,154]. Particularly under conditions of heating of milk between 65 and 85 °C, β-lactoglobulin complexes with MFGM proteins, presumably in response to unfolding of β-lactoglobulin (reviewed in [65]). Consistent with this finding, buttermilk-derived MFGM isolates contained large amounts of β-lactoglobulin, along with caseins and α-lactalbumin [154].

## 6. Relation between β-Lactoglobulin Fold and Digestion and the Gastro-Intestinal Immune System

Processing-induced changes of β-lactoglobulin in milk, such as unfolding, aggregation, and lactosylation, may impact digestion, adsorption, and the response of the gastro-intestinal immune system. For example, the extent of heating was observed to affect digestibility of proteins as a function of denaturation, aggregation, or lactosylation [155].

### 6.1. Protein Digestion and Absorption in the Gastro-Intestinal Tract—An Overview

Food-derived proteins are digested in the gastrointestinal tract. Susceptibility of proteins to digestion depends on isoelectric point [156] and structural organization. Once a dairy product arrives in the stomach, hydrochloric acid in gastric juices drive the pH in the gastric environment below 2, initiating the digestion of proteins mediated by the enzyme pepsin. Protein digestion involves the hydrolysis of peptide bonds, resulting in the formation of peptide fragments that progressively become shorter upon increased enzymatic processing. After partial digestion in the stomach into large peptide fragments, the majority of further protein digestion and absorption occurs in the duodenum by exposing the large peptide fragments to a mixture of pancreatic proteolytic enzymes chymotrypsin, carboxypeptidase, and trypsin to further hydrolyze peptide bonds, to produce smaller peptide fragments. Intestinal epithelial cells adhered brush border enzymes that include dipeptidases and amino peptidases and then hydrolyze the remaining peptide bonds further to generate even smaller fragments consisting of tripeptides, dipeptides, and individual amino acids. To aid absorption, the remaining tripeptides and dipeptides are taken up by the gastrointestinal epithelial cells by means of a cotransporter with hydrogen. Inside the cell, intracellular peptidases are active to hydrolyze any remaining peptide bonds and generate individual amino acids. Luminal individual amino acids are taken up by epithelial cells using a Na^+^ channel. These individual amino acids can then diffuse into the blood stream from which they are transported to the liver. While many enzymes used in in vitro digestion studies of β-lactoglobulin are from porcine origin, it was shown that human and porcine pepsin share similar specificity but differ in rate of activity [157].

### 6.2. Native β-Lactoglobulin is Largely Resistant to Gastric Digestion

The native β-lactoglobulin fold is subject to the so-called Tanford transition: a reversible conformational change that takes place in the EF loop (Figure 1B) when varying the pH between 6 and 8. While the pH of fresh milk is 6.5 to 6.75, upon ingestion and transport to the gastric cavity, milk proteins are exposed to a lowering in pH, giving rise to accumulation of a closed loop conformation of β-lactoglobulin [158]. Another pH driven conformational response of β-lactoglobulin to a lowering of pH is the dissociation of its two component dimers into monomers at pH values below 3 [159]. Prediction of cleavage sites in β-lactoglobulin upon exposure to gastric pepsin and trypsin in the duodenum leads to the identification of multiple sites susceptible to proteolysis by gastric pepsin (Figure 3). However, digestion of β-lactoglobulin upon exposure to gastro-intestinal enzymes is generally reported to be partial with a significant portion of the protein remaining intact after gastric digestion [160,161,162,163]. On the other hand, duodenal digestion, usually mimicked in vitro by exposure to pH 6.5, the enzyme trypsin, sometimes in the presence of pancreatic lipase and colipase, phosphatidyl choline vesicles, and bile salts, resulted in fragmentation of the protein with a low quantity of remaining intact protein [160,164,165,166,167,168]. In the presence of lipids β-lactoglobulin is more resistant against pancreatic proteolytic degradation, possibly as a result of its capacity to bind to lipid-like ligands stabilizing β-lactoglobulin conformation and limiting access to enzyme cleavage sites [160], similarly to α-lactalbumin [169]. Most of these observations were made under conditions of in vitro static digestion, and while such systems provide valuable information on the susceptibility of proteins to gastro-intestinal degradation, other factors, such as microbe-derived enzyme activity, are generally not taken into consideration. Digestion being a process with dynamic factors, including peristalsis, in vitro dynamic models for digestion are increasingly being developed and used to provide additional insight into protein digestion processes with the aim of more accurately recapitulating the in vivo digestion processes [170]. In a study comparing a static and a dynamic digestion protocol with in vivo digestion in a pig model, protein hydrolysis patterns appeared very similar in all three models, but while digestion endpoints were similar, kinetics of the hydrolysis process upon in vitro dynamic digestion more closely resembled those of the in vivo model [171].

### 6.3. Digestion and Absorption of Denatured and Aggregated Proteins in the Gastrointestinal Tract

Proteins with altered conformation as a result of processing may be subject to alternative digestion pathways or susceptibility. The effect of milk processing on digestibility has been systematically reviewed and denaturation was found to generally aid hydrolysis in the gastric cavity and affects gastric emptying (reviewed in [173]). Heat-mediated dissociation of dimeric β-lactoglobulin and unfolding leading to exposure of buried hydrophobic amino acids are considered to increase the accessibility to digestive enzymes such as pepsin [174,175,176]. While partial denaturation in the absence of extensive aggregation may increase susceptibility to enzymatic digestion as a result of exposure of (additional) cleavage sites, aggregation is generally thought to compact the protein structure to render such sites inaccessible. In line with this thought, it has been demonstrated that particularly the cores of fibrillar aggregates derived from prion proteins exert high levels of resistance against enzymatic breakdown [177]. The reported capacity of β-lactoglobulin to bind to lipid ligands and the suggested resistance of self-assembled β-lactoglobulin structures to proteolytic digestion have been exploited to investigate its use as an oral delivery system for improving the bioavailability of various hydrophilic and hydrophobic bioactive compounds, such as riboflavin and quercetin [178]. However, while heating of whey protein isolates was indeed shown to decrease pepsin-mediated digestibility [156], pure β-lactoglobulin fibrillar aggregates, generated upon heating at 80 °C and pH 2 or pH 7.4, were found to be rapidly digested by pepsin [179]. Similarly, heating of isolated β-lactoglobulin at 90 °C was shown to induce aggregation and increase in vitro gastric digestibility with differently sized aggregates exerting different degrees of susceptibility to proteolysis [168]. Interestingly, upon pepsin-mediated digestion in simulated gastric fluid, the generated peptide fragments were capable of re-association into newly formed fibrils [180,181]. It is possible that other constituents present in whey protein isolate, such as α-lactalbumin, bovine serum albumin, lactose, or lipids, are responsible for the observed limited susceptibility of aggregates to pepsin-mediated digestion which is also in line with the observation that β-lactoglobulin may form co-aggregates with other whey proteins (see Section 5) and that the presence of lipids decreases enzymatic digestibility of β-lactoglobulin [160]. While many studies investigate pepsin-mediated digestion of β-lactoglobulin, the trypsin-mediated digestion of β-lactoglobulin aggregates is relatively seldomly studied. While aggregated β-lactoglobulin was shown to be targeted by trypsin, when studied at a pH of 9.1, the rate of digestion was comparatively slow compared to native β-lactoglobulin and the generated fragments were incapable of reaggregation [181].

### 6.4. Digestion of Lactosylated β-Lactoglobulin

Pepsin was reported to preferentially cleave at phenylalanine, tyrosine, tryptophan, or leucine [182]. At the same time, prediction of cleavage sites using the ExPASy tool PeptideCutter [172] suggests that trypsin preferentially targets arginine and lysine groups [182], indicating that lysine modification may well affect the digestibility of proteins. Experimentally observed lysine lactosylation sites [75,76] also overlap with targeted cleavage sites for trypsin (Figure 3). It was indeed experimentally validated that lysine modification impairs susceptibility of proteins to enzymatic digestion (reviewed in [173,183,184,185,186,187,188,189]). Various mechanisms to explain impaired digestion have been suggested, including direct protection of lysine as a proteolytic target site, prevention of docking of proteolytic enzymes as a result of a modified lysine residue located immediately adjacent to an intended cleavage site, or by means of cross-linking with target residues [190].

## 7. Response of the Gastro-Intestinal Immune System to Heat-Treated β-Lactoglobulin

### 7.1. Mucus Layer as the First Line of Physical Defense

The immunological defense system of the gastrointestinal tract is composed of a mucosal layer that acts as first line of physical defense. The mucosal layer, which prevents the direct interaction of the gut epithelial layer with, for example, pathogenic bacteria, toxins, and other infectious agents, is composed of entangled forms of the glycoprotein mucin which is secreted by goblet cells. The gut’s resident microbiome and the mucus layer were shown to interact and support each other’s functioning by means of chemical interactions mediated by epithelial gene expression responses and innervation by the enteric nervous system. Some studies have investigated the diffusion of proteins and other molecules or particulates through the mucus layer. β-Lactoglobulin, and other proteins, can electrostatically interact with salivary mucins (reviewed in [191]), and bovine serum albumin, lysozyme, and α-synuclein proteins were observed to reorganize mucin MUC5B-containing hydrogels [192]. The rate of diffusion appears to be dictated, apart from by interaction with components of the mucus layer, by particle size, as the pore size of intestinal mucus was found to approximate 100 nm [193]. For example, it has been observed that mucus allows diffusion of 100 nm latex beads, and diffusion of 500 nm beads was limited [194]. As spherical aggregates of β-lactoglobulin were reported to vary between 100 and 1500 nm in size [109,110,111], it is plausible that diffusion of such particles is limited and a function of efficient proteolytic processing prior to adsorption.

### 7.2. Gut-Associated Lymphoid Tissue and Peyer’s Patches

Associated with the mucosal layer is the gut-associated lymphoid tissue (GALT) that is comprised of T and B lymphocytes, plasma cells, and macrophages organized into so-called Peyer’s patches covered with epithelial microfold, or M, cells in the small intestine. The basal membrane of M cells is extensively folded but lack apical microvilli, and these cells are responsible for uptake of antigens by means of endocytosis or phagocytosis, after which the antigen is encapsulated into transport vesicles and trafficked across the cell for release at the basal surface which is generally associated with a B cell [195] and dendritic cells [196]. Upon release, a mucosal immune response or mucosal tolerance may be provoked. M cells were shown to be a prerequisite for mucosal immune response, as an impaired mucosal immune response was observed in a model system deficient in M cells [197]. Peyer’s patch resident Th2 and Th3 cells respond to antigens by inducing their migration to the periphery and transforming growth factor (TGF)-β, interleukin (IL)-4, and IL-10 cytokine production [198,199,200]. The response of Peyer’s patches to luminal antigens appears to be regulated by pathogen recognition receptors of which Nucleotide oligomerization domain 2 (Nod2) and toll-like receptors have been established to play a key role (reviewed in [201]).

### 7.3. Uptake of β-Lactoglobulin by Microfold Cells

A few transmembrane capture receptors have been identified to play a role in uptake recognition by M cells, including the tight junction protein Claudin-4 [202] and glycoprotein 2 (gp2) [203], but currently none of those candidates have been specifically tested for recognition of β-lactoglobulin aggregates or other proteinaceous assemblies. At the same time, it is plausible to assume that uptake of β-lactoglobulin aggregates occurs via a similar pathway as has been previously identified for latex microparticles which were shown to be efficiently taken up by M cells before [204]. M cells were recently proposed to exert specialized uptake activity towards negatively charged microparticles [205], and both the diameter and electrostatic charge of β-lactoglobulin aggregates resemble those of such microparticles. Transcytosis of particular aggregates has remained an equally little-understood matter.

### 7.4. Response of Gastrointestinal Lymphocytes to β-Lactoglobulin

#### 7.4.1. Peyer’s Patch Lymphocytes

The response of Peyer’s patches to pathogenic bacteria and viruses has been well-described [206,207]. At the same time, limited studies investigated the response of Peyer’s patches to β-lactoglobulin. Exposure of human ileal Peyer’s Patches to β-lactoglobulin induced the proliferation and activation of Peyer’s patch resident lymphocytes, as indicated by increased expression of cluster of differentiation (CD)4^+^, CD8^+^, CD25^+^, C-C chemokine receptor type 5 (CCR5)—which is a Th1-associated chemokine receptor, and interferon (IFN)-γ [208]. In line with these observations, IL-10 and IL-4 secretions were not induced by dietary β-lactoglobulin showing that the response of Peyer’s patches from ileal origin to β-lactoglobulin is primarily Th1 type driven. Aggregation of proteins has been reported to modulate the intestinal immune response and Peyer’s patches were found to play a role too. While the precise uptake, transcytosis, and delivery mechanism of aggregated proteins by M cells has not been investigated, some studies reported on Peyer’s patch responses to various forms of β-lactoglobulin. While native β-lactoglobulin is primarily taken up through enterocytes [116], or, alternatively, was observed to drive a Th1 type response in Peyer’s patches [208], pasteurization-induced aggregation of β-lactoglobulin reduces uptake by absorptive epithelial cells, and fluorescein isothiocyanate (FITC)-labeling of β-lactoglobulin and imaging of Peyer’s patches using immunostaining showed that such aggregates are preferably taken up by Peyer’s patches both in vitro using monolayers of Caco-2 cells, which is a human epithelial colorectal adenocarcinoma cell line, and in vivo in a mouse model [116]. Uptake of aggregated β-lactoglobulin by Peyer’s patches was associated with an increased production of Th2-associated antibodies and cytokines such as IL-5, IL-13, and IFN-γ [116].

#### 7.4.2. Lamina Propria-Associated Lymphocytes

Apart from in Peyer’s patches, lymphocytes are also diffusely distributed in the underlying connective tissue called lamina propria. Similarly to Peyer’s patch resident lymphocytes, proliferation of lymphocytes associated with the jejunal lamina propria also was induced in vitro upon exposure to β-lactoglobulin [209]. β-Lactoglobulin-mediated proliferation of lamina propria CD4^+^ lymphocytes initially showed increased generation of IL-2, IL-10, IL-12, tumor necrosis factor (TNF)-α, and IFN-γ. IFN-γ induces the expression of surface markers on antigen presenting cells (APCs), which, in turn, generate the cytokine IL-12. IFN-γ has been shown to induce IFN-γ production in a CD2 signaling pathway-mediated manner [210]. The involvement of APCs was demonstrated as antibodies against CD2 or its ligand CD58 reduced β-lactoglobulin-mediated proliferation of lymphocytes. At the same time, prolonged exposure to β-lactoglobulin triggered suppressor activity [209]. While it is clear that native β-lactoglobulin modulates the proliferation and cytokine expression patterns of lymphocytes, whether and how lamina propria-resident lymphocytes respond to aggregated β-lactoglobulin has not been reported.

### 7.5. Human Leukocyte Response to β-Lactoglobulin after Absorption

Neutrophils are blood-residing polymorphonuclear leukocytes that are recruited to the intestinal lamina propria in response to a chemoattractant gradient generated by the monocytes present in the gut [211]. Once trafficked to the lamina propria or upon transepithelial migration into the intestinal lumen, neutrophils play a role in the gastrointestinal immune response by generating reactive oxygen species, cytokines, and antimicrobial peptides; eliminating pathogenic bacteria; and recruiting macrophages by secreting monocyte chemoattractants (reviewed in [212]), [213]. While the response of neutrophils to invading pathogenic bacteria has been well-established, reactivity toward whey proteins and aggregates thereof is less well known. β-Lactoglobulin was found to induce the generation and secretion of self-regulatory interleukin receptor 1 antagonist by neutrophils in vitro mediated by p38 MAPK, MAPK/ERK kinase, and NF-κB pathways, and to stimulate the neutrophilic production of inflammatory mediators IL-1β, IL-8, IL-6, TNF-α, and macrophage inflammatory proteins 1α and 1β [214]. β-Lactoglobulin was also found to induce neutrophilic superoxide production and induce translocation of p47phox subunit of cytosolic NADPH-oxidase to the plasma membrane [215] which is required for activation of NADPH-oxidase-mediated superoxide production [216].

## 8. Conclusions

β-Lactoglobulin has been subject to intensive investigation efforts since its identification as a main whey protein component. While the biological function of this protein remains to be revealed, β-lactoglobulin was shown to be a highly complex protein molecule able of undergoing various transitions, such as the pH-induced Tanford transition, thiol-disulfide reshuffling, and dimer–monomer conversion, while, at the same time, being able to populate various transient states such as the molten globule and various aggregated forms. β-Lactoglobulin is also subject to various types of modifications, such as lactosylation, and it is reported to interact with a number of other proteins present in milk. Many of these reported features of β-lactoglobulin are now known to be altered, to some degree, mediated by homogenization and heat-processing of milk aimed at extending stability and shelf-life of the perishable product. In turn, the different modifications β-lactoglobulin undergoes upon thermal processing have implications for the digestibility, bioavailability, and gastrointestinal response to β-lactoglobulin. These multi-facetted, dynamic, and transient short-lived characteristics of the various faces β-lactoglobulin can assume complicate the study of its physico-chemical and biochemical properties. In many instances, it turned out to be challenging, in part because of the dynamics of states, to unequivocally associate any of the reported biophysical and biochemical characteristics of β-lactoglobulin to biological responses upon ingestion. To address that, many researchers to date have succeeded in revealing some of the complex interactions and structural-biological features of β-lactoglobulin by working with isolated β-lactoglobulin fractions, away from the complex environment that liquid milk represents. In the future, we envision that studies into the behavior of β-lactoglobulin in milk may reveal further molecular insights in the workings of this multi-functional protein.

## Figures and Tables

**Figure 1 foods-09-00874-f001:**
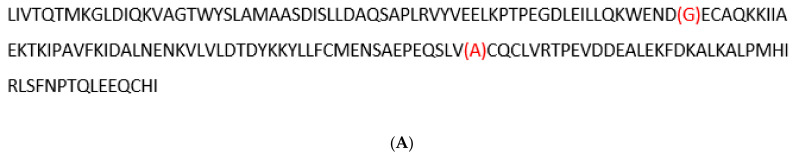
Structural features of bovine β-lactoglobulin. (**A**). Primary sequence of β-lactoglobulin variant A (B). (**B**). Diagram of secondary structural features of β-lactoglobulin. H1–H4, helical segments; A1–I, beta sheets; β, beta turn; γ, gamma turn; ①=②; disulphide bonds. Figure generated using PDBsum, PDB ID 1BSO [30], (**C**). Crystal structure of monomer of bovine β-lactoglobulin monomer, PDB ID: 1BEB [28]. Image generated using RCSB PDB [31].

**Figure 2 foods-09-00874-f002:**
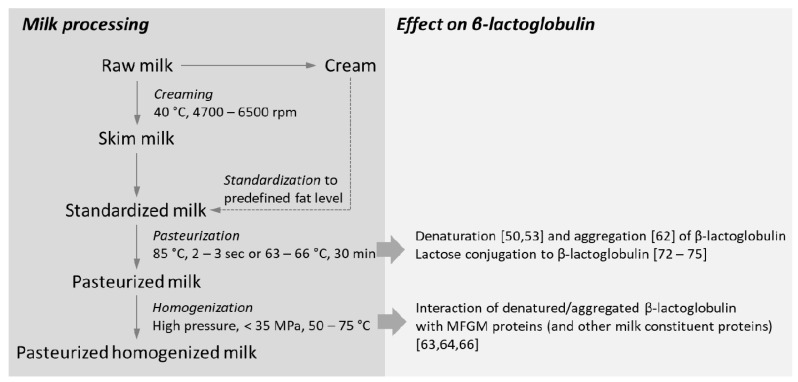
Processing of milk involves various steps, including creaming, pasteurization, and homogenization, which may impact β-lactoglobulin conformation and interaction with milk constituent proteins. The numbers in this image refer to publications in the reference list.

**Figure 3 foods-09-00874-f003:**
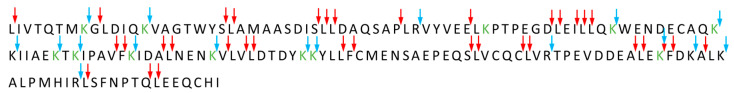
Predicted cleavage sites of pepsin at pH 1.3 (red arrows), mimicking gastric digestion, and trypsin (blue arrows), mimicking intestinal digestion. Cleavage sites were predicted using the amino acid sequence of β-lactoglobulin A as input in ExPASy, using the tool PeptideCutter [172]. Lysine (K) residues indicated in green have previously been identified to be subject to lactosylation [75,76].

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
