# Peer review of "Milk Processing Affects Structure, Bioavailability and Immunogenicity of β-lactoglobulin"

_foods, 2020, doi:10.3390/foods9070874_

Round 1
Reviewer 1 Report
Dear Authors,
The issues discussed in the article are very current due to the high consumption of cow's milk. Prior to consumption, raw milk directly derived from the animal is subjected to various processing steps that may affect its molecular constituents.
The biological function of β-lactoglobulin remains under discussion.
I believe that this protein fraction has great potential and in the light of the presented facts should still be the subject of many studies in this area.
Literature is well-chosen, but only 24% of the last 10 years.
Author Response
Reviewer 1.
English language and style are fine/minor spell check required.
The manuscript has been thoroughly checked for spelling and grammar and a few changes have been made. I refer to track changes to identify the changes made.
Literature is well-chosen, but only 24% of the last 10 years.
It is interesting that this reviewer notices the same trend as I did myself during my literature survey. Many references studying β-lactoglobulin are indeed less recent and were originally focused on structural and biological features of this protein. While searching for publications related to this review I noticed that, while the number of publications referring to β-lactoglobulin in Pubmed have increased with the years, more recent works on β-lactoglobulin more frequently focus on technological aspects of this protein, such as sensing, as biomedical engineering solution, and as a drug carrier or as active ingredient, such as foaming. As these topics are not in scope with the current review they were generally omitted.
Reviewer 2 Report
This article is quite important as it makes a thorough review of one of the main whey proteins, ? − lactoglobulin, with high nutritional value. A large number of references have been consulted (218), and it can be seen that a large part of them correspond to research carried out since the sixties of the last century, with a higher concentration of references in the nineties. This trend has continued into the 21st century, where intensive research continues to be conducted with the aim of understanding the influence of the main pre-treatments of milk, namely homogenization and heat treatments, on the structure of this protein and consequently its behaviour in the gastro intestinal immune system.
However, as the author herself points out, most studies have been carried out with pure protein solutions, so that more studies in real samples, either in milk or in its whey, are needed in order to obtain more consistent results. Getting results closer to reality will help in the design and development of more precise dynamic models "in vivo" of the digestion processes of this protein.
I want to suggest some minor corrections of this work:
- In figure 2, it is important that the author indicates if the ?-lactoglobulin changes presented belong to some of the references consulted and which of them, or if they result from experimental work performed by the author herself? For example, in the low pasteurization conditions mentioned, several authors mention that few or no changes occur in whey proteins.
- The author should make the following spelling corrections:
L55 - replace "biology role" with "biological role.
L397 - point 6.3 should be 6.2, as it follows 6.1.
Author Response
Reviewer 2.
This article is quite important as it makes a thorough review of one of the main whey proteins, ? − lactoglobulin, with high nutritional value. A large number of references have been consulted (218), and it can be seen that a large part of them correspond to research carried out since the sixties of the last century, with a higher concentration of references in the nineties. This trend has continued into the 21st century, where intensive research continues to be conducted with the aim of understanding the influence of the main pre-treatments of milk, namely homogenization and heat treatments, on the structure of this protein and consequently its behaviour in the gastro intestinal immune system.
However, as the author herself points out, most studies have been carried out with pure protein solutions, so that more studies in real samples, either in milk or in its whey, are needed in order to obtain more consistent results. Getting results closer to reality will help in the design and development of more precise dynamic models "in vivo" of the digestion processes of this protein.
I want to suggest some minor corrections of this work:
- In figure 2, it is important that the author indicates if the ?-lactoglobulin changes presented belong to some of the references consulted and which of them, or if they result from experimental work performed by the author herself? For example, in the low pasteurization conditions mentioned, several authors mention that few or no changes occur in whey proteins.
Adding references to this figure would indeed be helpful so readers can refer back to this information in an actual experimental context. References have now been added to figure 2 and the figure legend has been adapted.
- The author should make the following spelling corrections:
L55 - replace "biology role" with "biological role.
This has been changed, see page 2 track changes in the manuscript.
L397 - point 6.3 should be 6.2, as it follows 6.1.
This has been changed. See page 12 track changes in the manuscript. The follow-up sections have also been renumbered consequently.